# Insight into the Antifungal Effects of Propolis and Carnosic Acid—Extension to the Pathogenic Yeast *Candida glabrata*: New Propolis Fractionation and Potential Synergistic Applications

**DOI:** 10.3390/jof9040442

**Published:** 2023-04-04

**Authors:** Alejandra Argüelles, Ruth Sánchez-Fresneda, José P. Guirao-Abad, José Antonio Lozano, Francisco Solano, Juan-Carlos Argüelles

**Affiliations:** 1Vitalgaia España S.L., 30005 Murcia, Spain; 2Área de Microbiología, Facultad de Biología, Universidad de Murcia, 30071 Murcia, Spain; 3Departamento de Bioquímica y Biología Molecular B e Inmunología, Facultad de Medicina, Universidad de Murcia, 30120 Murcia, Spain

**Keywords:** natural antifungal, *Candida glabrata*, *Rosmarinus officinalis*, *Apis mellifera*, propolis, hydroethanolic fractionation, HPLC, amphotericin B

## Abstract

Fungi have traditionally been considered opportunistic pathogens in primary infections caused by virulent bacteria, protozoan, or viruses. Consequently, antimycotic chemotherapy is clearly less developed in comparison to its bacterial counterpart. Currently, the three main families of antifungals (polyenes, echinocandins, and azoles) are not sufficient to control the enormous increase in life-threatening fungal infections recorded in recent decades. Natural substances harvested from plants have traditionally been utilized as a successful alternative. After a wide screening of natural agents, we have recently obtained promising results with distinct formulations of carnosic acid and propolis on the prevalent fungal pathogens *Candida albicans* and *Cryptococcus neoformans*. Here, we extended their use to the treatment against the emerging pathogenic yeast *Candida glabrata*, which displayed lower susceptibility in comparison to the fungi mentioned above. Taking into account the moderate antifungal activity of both natural agents, the antifungal value of these combinations has been improved through the obtention of the hydroethanolic fractions of propolis. In addition, we have demonstrated the potential clinical application of new therapeutical designs based on sequential pre-treatments with carnosic/propolis mixtures, followed by exposure to amphotericin B. This approach increased the toxic effect induced by this polyene.

## 1. Introduction

Historically, fungi have been considered less relevant pathogenic organisms than bacteria or viruses for the search and application of biomolecules with chemotherapeutic purposes. In accordance, the investigation of antimycotic antibiotics has attracted less interest from both the public system and the pharmaceutical industry. A crucial problem derives from the fact that fungi are eukaryotic organisms and infect other eukaryotic hosts with an identical or, at least, very similar cellular organization. This close cellular resemblance concerning cell structure, functional processes, and metabolic pathways has made the task of disturbing specifically the fungal cell somewhat difficult.

Nevertheless, this scenario has changed in recent decades with the large increase in morbidity and mortality caused by fungal infections and the isolation of several species of fungi as responsible for life-threatening outbreaks recorded in hospitals and in the community [1,2,3,4,5,6,7]. The highest prevalence corresponds principally to the genera *Candida*, *Aspergillus*, and *Cryptococcus*, and several dimorphic fungi [1,4,6,8,9]. Likewise, the growing spread of clinical strains resistant to conventional antifungals complicates the development of safe and efficacious chemotherapy against opportunistic infective fungi [2,5,10,11,12].

The antifungal arsenal available for clinical therapy is composed of three main families of compounds: polyenes, azoles, and echinocandins, plus some agents with a different chemical structure and mechanism of action (i.e., Griseofulvin, Terbinafine, or Ciclopirox) [11,13,14]. Most of them were introduced for biomedical applications 50–60 years ago, and their use remains almost invariant [15]. Thus, the polyene amphotericin B (AmB) is still widely applied against many prevailing mycoses, although it displays some toxic side effects on the liver and kidney (controlled by novel liposomal formulations) [16]. Furthermore, recent data point out that the mechanisms involved in antifungal treatments and, therefore, the pathways activated by pathogenic fungi to achieve resistance to those agents, are more complex than initially thought [17,18,19]. In quotidian medical practice, this stock is clearly insufficient to combat both superficial and systemic mycoses [11,12,13].

Collectively, this evidence indicates that more input and efforts are necessary to obtain, characterize, and improve new molecules to increase the arsenal of useful compounds endowed with antimycotic activity. Undoubtedly, the main preferential targets should be the structural components of fungi or key proteins with an essential role in cellular viability. There are many plants that contain bioactive agents able to impair metabolic pathways [2,5,20]. In this context, natural substances obtained from plants and other natural sources have certainly been known since ancient times for their use in folk medicine, and some of them have been incorporated into antimicrobial therapy already. In other cases, some targeted investigations have identified new substances that afford promising expectations. We have obtained successful in vitro synergistic results through specific combinations of carnosic acid (CA) and propolis (PP) against opportunistic yeasts [21,22]. CA is a benzenediol diterpene extracted from rosemary (*Rosmarinus officinalis*) leaves [23], and PP is a commercially available complex mixture produced by honeybees (*Apis mellifera*) [24,25]. Both are antioxidant natural agents with several biotechnological applications and beneficial effects due their antioxidant and anti-inflammatory properties [26,27,28]. Here, the validity of this antifungal synergy has been extended to the emerging pathogenic yeast *C. glabrata* [29,30,31,32]. We also document that the ethanolic fractionation of propolis could improve the fungicidal effects of future CA:PP formulations. In addition, the potential therapeutic application of new combinations between these natural extracts and well-established antifungals has also been examined.

## 2. Materials and Methods

### 2.1. Yeast Strains and Culture Conditions

The standard strain *C. glabrata* ATCC 2001 and the oral clinical strain *C. albicans* 015 were used throughout this study. The clinical isolate from Hospital “La Fe”, Valencia, Spain, was a gift from Prof. E. Valentín. Yeast cell cultures were grown at 37 °C by shaking in YPD medium consisting of 2% peptone, 1% yeast extract (Condalab, Barcelona, Spain), and 2% glucose (Merck, Darmstadt, Germany). The strains were refreshed from a −80 °C glycerinated stock and maintained at 4 °C by periodic subculturing in solid YPD containing 2% agar or Sabouraud dextrose medium.

### 2.2. Determination of Cell Viability

Identical aliquots of the above-indicated yeast strains grown at 37 °C in liquid YPD until they reached the exponential phase (OD_600nm_ = 0.8–1.0) were treated with the indicated concentrations of CA and PP. For the combinations between CA:PP mixtures and conventional antifungals, growing cultures of *C. albicans* 015 were pre-treated with CA:PP for 1 h at 37 °C, and AmB (0.05 or 0.1 μg/mL) was added immediately. Then, the incubation was maintained for another hour. Cell viability was determined in samples diluted appropriately by plating in triplicate on solid YPD after incubation for 1–2 days at 37 °C. Survival percentages were normalized to control samples (100% viability). The statistical significance of the differences was determined using the Mann–Whitney U test. Colony growth in solid medium was tested by spotting 5 μL from the respective ten-fold dilutions onto YPD agar. Then, the plates were incubated at 30 °C and scored after 48–72 h.

### 2.3. Natural Sources and Ethanolic Fractionation of PP in Solid Columns

Semi-purified CA was obtained from fresh leaves of rosemary (*Rosmarinus officinalis*) by conventional solid–liquid extraction as reported elsewhere [33]. Purity (around 75%) was verified by HPLC connected to a Diodo-Array Agilent UV–Vis detector [22]. The extract was dissolved in ethanol up to 1 mg/mL and diluted until the required concentration was reached for the antifungal assays.

The raw PP was obtained from local suppliers. These extracts were originally collected in China. The raw PP was then ground to a fine powder, dissolved by gentle shaking in prewarmed ethanol, and filtered according to Arguelles et al. [21,22].

The fractionation of crude dissolved PP was performed in full cartridges Strata-X (33 µm polymeric reverse 60 mg/3mL 8B-S100-UB) SPE (Phenomenex) according to the following protocol: the cartridge was initially equilibrated by the subsequent addition of methanol (2 mL) and ammonium acetate 20 mM (2 mL). After that, 200 µL of the crude PP extract (500 μ/mL) was loaded onto the column and, immediately, another 200 µL of ammonium acetate 20 mM was added for equilibration. Then, the cartridge was washed with 2 mL of 20 mM ammonium acetate solution containing 15% methanol, incubated at room temperature for 10 min, and the eluate was collected after soft pulse (1000 rpm, 15 s). The cartridge was then successively loaded with 200 µL of solution containing growing concentrations of ethanol, and the eluates were recovered as indicated. Identical small aliquots from each fraction were taken for HPLC analysis (see Section 2.4). Alternative aliquots were used for the determination of the antifungal activity and gravimetric quantitation. Hydroethanolic fractions were gently evaporated (600 mm Hg pressure, 60 °C) in a vacuum rotavapor until they reached total dryness and were weighed. Then, the solids were gently dissolved in ethanol for antifungal assays, per se or in combination with CA. Total mass recovery was around 57% due to loss as cartridges retained substances or volatile compounds, and the EtOH75 active fraction contained around 25% of the crude PP weight.

### 2.4. HPLC

The conditions for analytical HPLC were described elsewhere [34] with slight modifications. Sample hydroalcoholic fractions of PP (5 mg/mL) were dissolved in DMSO and filtered through a nylon membrane with a 0.45 mm pore size. Then, 20 µL of solution was injected in a LiChrospher 100-C18 reverse-phase column (250 × 4.0 mm inner diameter) thermostatized at 30 °C. The mobile phase consisted of a gradient of acetonitrile/2.5% acetic acid aqueous solution, starting with 5%/95% and finishing with 95%/5% acetonitrile/acidic water. The flux was 1 mL/min. Detection was followed with a DAD-UV-Vis detector, as described above for CA analysis. Usually, chromatographic profiles were monitored at 280 and 340 nm for an approximate identification of phenols and flavonoids, but the complete spectra of some selected major peaks could be analyzed for the identification of the selected components.

## 3. Results and Discussion

### 3.1. The Synergistic Fungicidal Effect of CA and PP Can Be Extended to Other Emerging Yeasts

Conspicuous antifungal activity induced by precise combinations of CA plus crude PP has been demonstrated against the prevalent pathogenic yeasts *Candida albicans* (ascomycete) and *Cryptococcus neoformans* (basidiomycete) [21,22]. Distinct CA:PP ratios were tested, and the strongest fungicidal action was recorded at a 1:4 (CA:PP) proportion [21]. A mathematical analysis by means of the application of isobolograms and a quantitative determination of the fractional inhibitory concentrations index (FICI) strongly support that the positive cooperation of both natural extracts takes place through a synergistic action rather than a merely additive one [21,22,35].

We decided to extend this analysis to the opportunistic yeast *C. glabrata*, which has emerged as a fungus responsible for a significant percentage of invasive candidiasis in many countries of Europe and America [29,30,31,32]. *C. glabrata* bloodstream infections usually cause a high mortality rate because this yeast frequently contains a number of virulence factors, including intrinsic antifungal resistance [29,30]. In fact, *C. glabrata* displays increasing cross-resistance to azoles and echinocandins [7]. For comparative purposes, we have followed the same experimental approach previously used with *C. albicans* and *C. neoformans* [21,22]. As shown in Figure 1, the percentage of cell death recorded in either liquid YPD medium after 5 h upon the addition of the individual agents was moderate, although in *C. glabrata*, the toxic effect of the PP extract was quite similar to that recorded for CA. As expected, the largest toxic action was observed in the presence of a CA:PP (1:4) mixture. However, the capacity of *C. glabrata* cells to withstand these treatments was superior to that observed in other pathogenic yeasts at identical times of incubation [21,22].

Therefore, the assays were repeated including a new sample (S’), where the CA:PP ratio was maintained (1:4), but the concentration of both agents was duplicated. As a result, cell viability in the S’ treatment decreased to levels near 0.01% (Figure 2), which are like the values measured in other opportunistic fungi with individual compounds or a standard mixture [21,36,37]. Furthermore, the fractional inhibitory concentration index (FICI) determined from the corresponding MICs (see below) was 0.43, confirming that the recorded antifungal activity induced by CA:PP formulations is synergic (FICI ≤ 0.5) [35].

In turn, after the addition of a double concentration of each compound, the antifungal effect of the PP extract in planktonic cells (YPD liquid) was slightly higher than that displayed by CA (Figure 2A). These series focused on short times because of the elevated concentrations of both natural agents tested. Under those conditions, longer incubations (5 h) produced the appearance of a brownish color in the medium, indicating an oxidation of some active principles in the CA:PP mixture, probably the quinone form of CA. Therefore, the incubation was not prolonged beyond 2 h. The recorded pattern was different to that reported for *C. albicans* [21], suggesting a distinctive immediate sensitivity of both *Candida* species to these agents when added separately. However, the higher fungicidal effect of PP in comparison to CA was not maintained after a long time (48 h) in colonial growth (YPD solid), where CA induced a somewhat larger cell toxicity than the crude PP extract (Figure 2B).

The previously calculated MIC_50_ values for the two agents in the mixture following the XTT method were 62.5 μg/mL (CA) and 125 μg/mL (crude PP) on exponentially growing cultures of *C. neoformans* and *C. glabrata*, whereas for *C. albicans,* the obtained data were 62.5 μg/mL (CA) and 62.5 μg/mL (PP) [21,22]. Of course, these values are significantly elevated compared with the MICs recorded for clinical antifungals, and this fact would preclude their introduction in chemotherapy. However, it should be kept in mind that those studies were performed using semi-purified preparations of CA (70–75%) and crude extracts of PP, which is a heterogenous raw material with a variable chemical composition that includes more than 300 substances and is largely dependent on many biotic and abiotic factors, as stated above [24,25]. As a matter of fact, we have tested different sets of PP from distinct origins and observed that some of them are totally unable to induce any fungicidal effect, alone or in combination with CA [21,22]. For these reasons, an alcoholic fractionation of the whole active raw PP was carried out later, as described in Section 3.3. Furthermore, the active components within PP responsible for the observed synergistic antifungal actions are currently under identification from the active ethanolic fractions. At present, these data are under the industrial property of Vitalgaia, S.A., and intended for a future patent. However, it is expected that their future public knowledge will likely allow the determination of lower MICs and will further improve the fungicidal action recorded by new CA and PP formulations enriched in these active components.

Although the mechanisms accounting for the antifungal synergistic action of CA and PP have not been completely elucidated yet, it is well established that the inclusion of hydroalcoholic formulations improve their antimicrobial effects [25,36,38]. In the case of *C. neoformans*, the ethanolic fractions of PP diminished the content of chitin/chitosan involved in cell wall integrity as well as melanin production, which is a main factor of virulence [38,39]. In turn, the evidence gathered from *C. albicans* is weaker, but the fungicidal effect mainly relates to the antioxidant activity and the ability to scavenge the endogenous ROS generated during respiratory metabolism. In this regard, the synergistic action might be caused by an impairment of the inner redox balance triggered by the presence of CA and PP [21,22], although the antioxidant capacity of distinct PP can vary as a function of its heterogeneous chemical composition [24,25].

### 3.2. Potential Fungicidal Activity Achieved by Combination of CA-PP Mixtures and A Conventional Antifungal, Amphotericin B

Another interesting line of research deals with the hypothetical positive cooperation between established clinical antifungals and natural substances. In this context, synergistic action between Polish PP and two azoles (Fluconazole and Voriconazole) was effective in the eradication of preformed biofilms [40], whereas the combination of AmB with eugenol or with a set of off-patent drugs caused synergism when applied against some pathogenic yeasts [35,41]. Furthermore, it has been reported that other mixtures between PP and antibacterial antibiotics or antifungals have also shown a synergistic effect [42].

We carried out a set of assays by utilizing an oral isolate of *C. albicans* (015) previously tested in studies on CA:PP action [21]. The preliminary experiments were inconclusive, since the strong toxic effect induced by AmB (0.5 μg/mL) masked the results obtained with the mixture between this polyene plus the standard combination CA:PP (1:4). To circumvent this trouble, an alternative method was followed, in which growing cells were pretreated for 1 h with the CA:PP formulation at an identical ratio (1:4), but the concentration was reduced to half to prevent drastic cell damage. Then, the cultures were immediately exposed to lower concentrations of AmB (0.05 and 0.1 μg/mL) for another hour. This strategy has been functional for achieving thermotolerance and protection against severe oxidative stress in *Saccharomyces cerevisiae* and *C. albicans* [43,44].

The data presented in Figure 3 point to a higher level of cell death caused by the synergic mixture (CA:PP) compared to that recorded after the addition of two individual doses of AmB. However, the previous treatment with CA:PP clearly increased the further lethal activity triggered by the polyene. The assays regarding colonial formation on the solid YPD medium corroborate this result (Figure 3B). Notably, similar assays using micafungin instead of amphotericin B failed to produce an equivalent antifungal effect (data not shown). Therefore, and although more extensive experiments are necessary, our initial work predicted that the preincubation of active cells from prevalent pathogenic yeasts with the standard 1:4 (CA:PP) formulation, followed by a further exposure to AmB, should decrease its MIC values. Remarkably, this therapeutic approach seems convenient since mixtures with the inclusion of two natural agents could minimize the hepatic and nephrotoxic side effects displayed by the conventional preparations of AmB.

### 3.3. Hydroalcoholic Fractionation of Raw Propolis

The hydroalcoholic preparations of PP are customarily introduced in the design of new therapeutic formulations with antimicrobial activity. Hence, with the goal of exploring the nature of the fungicidal activity contained in the raw PP extracts [21,36], the samples of PP were subjected to ethanolic fractionation due to the complexity of that product. For this purpose, we used cartridges with hydrophobic packing followed by a stepwise elution with hydroethanolic mixtures containing a rising content of ethanol (EtOH) from 0 to 100% with increments of 25%. This solvent was chosen due to the friendly properties of hydroethanolic mixtures, and because it has previously been reported that ethanolic extracts of stingless bee PP enhance the strong fungicidal effect against *C. neoformans* through the reduction in the chitin–chitosan and melanin content [38,39].

The concentrations of the resulting fractions were determined by gravimetry after solvent evaporation (Section 2.3). Data are summarized in Table 1. It can be seen that most of the crude PP material shows hydrophobic features as it is only eluted in the EtOH75 and EtOH100 fractions. The main active components eluted in the EtOH75 fractions showed UV-Vis spectra consistent with flavonoids, especially the peak with a retention time of 42.64 min (Figure 4), whereas phenols seemed to be absent. The weight of the non-volatile components present in those fractions comprised around 50% of the total components found in the raw extracts, and only about 7% were eluted in the initial aqueous-rich fractions. To complete the balance of the PP material, a significant amount of the PP material (around 43%) was retained in the Strata-X cartridges or evaporated during the preparation of the fractions.

Appropriate aliquots of the eluted fractions were submitted for HPLC analysis with a DAD-UV-Vis detector to determine their composition. A representative chromatogram corresponding the EtOH75 fraction is shown in Figure 4, and the chromatograms obtained from the rest of the fractions are presented in Appendix A.

The antifungal activity of those fractions expressed as cell survival (%) and normalized to the respective concentrations was also assayed. As can be observed in Figure 5, the EtOH75 fraction displayed strong antifungal activity against the growing cells of the clinical strain 015 of *C. albicans.* This toxic effect was considerably higher compared to that recorded after treatment with crude extract. However, the other fractions eluted with just water, EtOH25, EtOH50, and EtOH100 were devoid of significant fungicidal activity.

Thus, the principal components responsible for the antifungal actions contained in the original PP extracts are related to the peaks contained in the EtOH75 fraction. The chemical identification obtained from the UV-Vis spectra suggests that the most prominent peaks eluted in the EtOH75 are flavonoids, whereas the predominant components eluted in the EtOH100 fractions are terpenoids and large-molecular-size substances. As stated above, the main active principles responsible for the antifungal action induced by propolis have also been identified. Presently, however, these data are under the industrial secret of Vitalgaia, S.A and cannot be published.

One interesting pending aspect of this study concerns the mechanisms of action involved in the observed antifungal effects. The targets of some clinically relevant compounds are either the plasma membrane or the cell wall of the fungal cell, disturbing the membrane permeability or the external network of structural exopolysaccharides. Thus, it is well known that echinocandins (e.g., micafungin) inhibit the β-(1,3)-D-glucan synthase that catalyzes the formation of β-glucan polymers, essential components of the cell wall, whereas polyenes (e.g., amphotericin B) interact with ergosterol present in the cell membrane to form ion channels that disrupt membrane function [45]. However, it has been demonstrated that AmB also causes oxidative damage in *C. albicans* [17,18]. Concerning the CA+PP mixtures, our preliminary experiments point out a relationship between those agents and the impairment of internal redox equilibrium mediated by the disruption of the mitochondrial function [21] together with the inhibition of cytosolic quinone reductases. Both agents (CA and PP) show antioxidant properties and have been proposed as positive supplements for cellular metabolism due to their capacity to scavenge ROS [5,23,27]. In spite of these beneficial effects of each antioxidant separately, the redox cellular balance is complex and intriguing. In fact, some reports challenge the widely assumed scientific belief that antioxidant agents are always beneficial for living cells [46]. At present, we cannot provide definitive evidence on the molecular mechanisms implied in either the antifungal effectiveness of CA+PP mixtures or those formulations combined with AmB. However, the analysis of the synergistic effects strongly suggests that several targets should cooperate to induce the remarkable antifungal action observed.

## 4. Conclusions and Perspectives

Ancient and current alternative medicine involves the empiric utilization of crude or semi-purified preparations obtained from plants and other sources as valuable remedies against symptoms and diagnosticated diseases. Careful scientific studies carried out in recent decades have permitted the identification of some responsible active components, followed by a further characterization of their biological features. Antimicrobial research targeting bacteria has prioritized fungi, although considerable progress on the obtention of natural extracts endowed with fungicidal activity has been achieved in recent years.

This contribution reinforces and extends the potential interest of two natural extracts, CA and PP, as an alternative therapeutical target to conventional antifungals. The evidence collated demonstrate that a few precise combinations of these bioactive extracts convey a robust fungicidal effect against invasive ascomycetes (*C. albicans* and *C. glabrata*) and basidiomycetes (*C. neoformans*), whereas the experimental data regarding filamentous fungi (*Aspergillus flavus*) are too preliminary. Notably, the action induced by CA:PP combinations is synergistic and not merely additive, and it can be largely improved by means of the alcoholic fractionation of crude PP. Moreover, its application joined to conventional antifungals warrants further investigations in order to elucidate two important matters: (i) the mechanisms of action involved in the synergistic antifungal activity triggered by CA:PP mixtures and (ii) the search for new interesting therapies based on alternative mixtures or the design of distinct combinations of natural substances with synthetic clinical antifungals, such as AmB.

## Figures and Tables

**Figure 1 jof-09-00442-f001:**
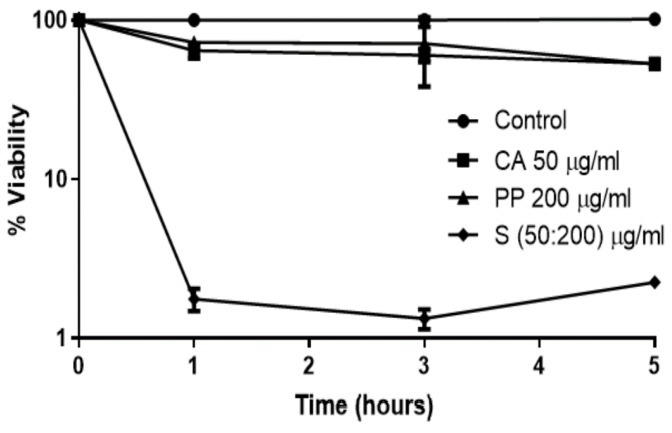
Time-course of cell killing (%) recorded in YPD-grown exponential cells of the standard *C. glabrata* ATCC 2001 strain, after the individual addition of 50 μg/mL of CA and 200 μg/mL of raw PP or a combination of the two natural agents (S for CA:PP at the indicated concentrations). After incubation at 37 °C, samples were harvested at the indicated times, washed, and the percentage of cell survival was determined by CFU counting. The plotted data shown are the mean ± SD of three independent measurements.

**Figure 2 jof-09-00442-f002:**
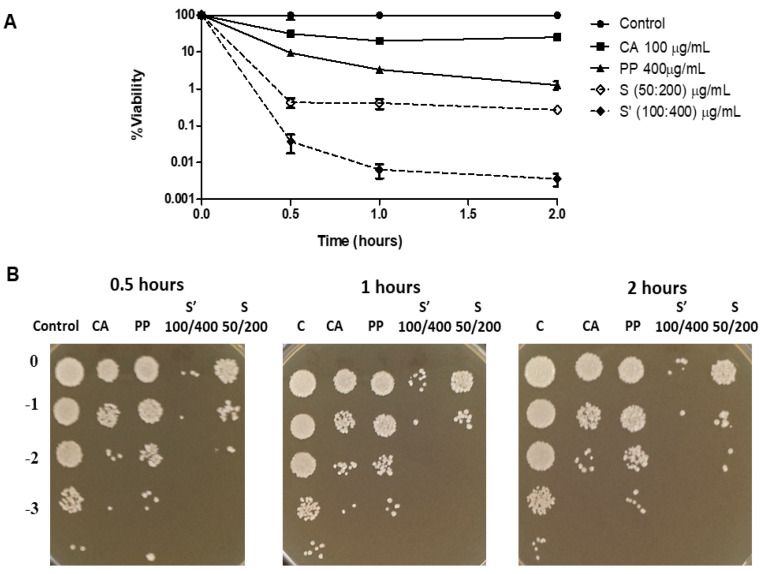
Level of cell viability in liquid YPD medium (**A**) and the formation of macroscopic colonies on solid YPD plates at 37 °C (**B**) of *C. glabrata* cells upon exposure with double concentration of CA (100 μg/mL) and PP (400 μg/mL). A mixture containing both agents (S’) was also included, and the original CA:PP treatment (S) was maintained as reference. Two different series were carried out with very similar pattern. (**A**) Data shown correspond to the mean ± SD of those series, and (**B**) is a picture of the colonies observed by dilution of one of the series.

**Figure 3 jof-09-00442-f003:**
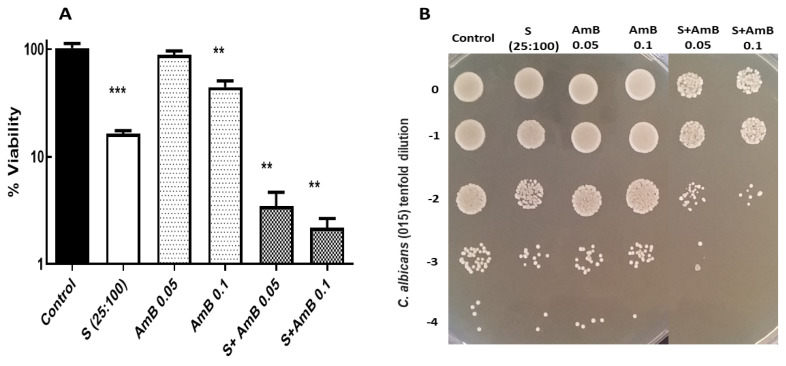
Antifungal effect induced by the combinative treatment with CA:PP mixture (S, 25:100 μg/mL), followed by AmB exposure on *C. albicans* 015 strain. YPD cultures recovered in exponential phase were preincubated for 1 h at 37 °C and immediately exposed to AmB (0.05 and 0.1 μg/mL) for another hour. The percentage of cell survival in liquid medium (**A**) is expressed with respect to an identical control untreated sample (100%) according to the Mann–Whitney U test. Statistically significant differences were: ** *p* < 0.01; *** *p* < 0.001. The colonial growth was recorded on solid YPD plates (**B**) incubated at 37 °C and scored after 48 h.

**Figure 4 jof-09-00442-f004:**
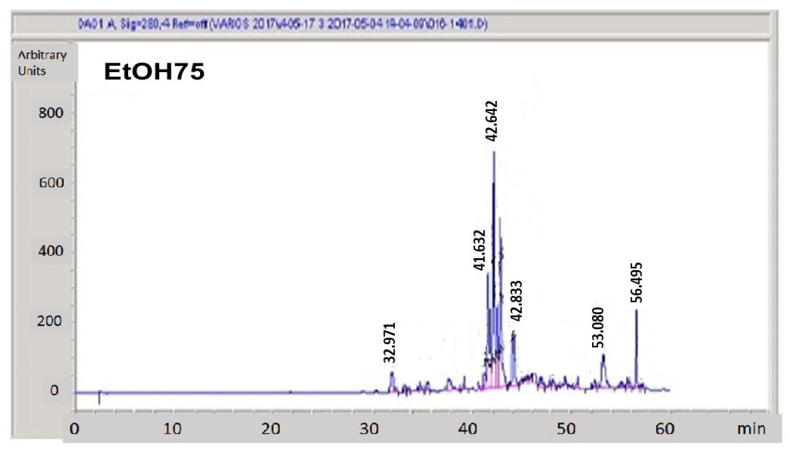
HPLC chromatogram of the EtOH75 fraction monitored by absorbance at 280 nm. For details on the hydroethanolic PP fractionation, see Methods Section 2.

**Figure 5 jof-09-00442-f005:**
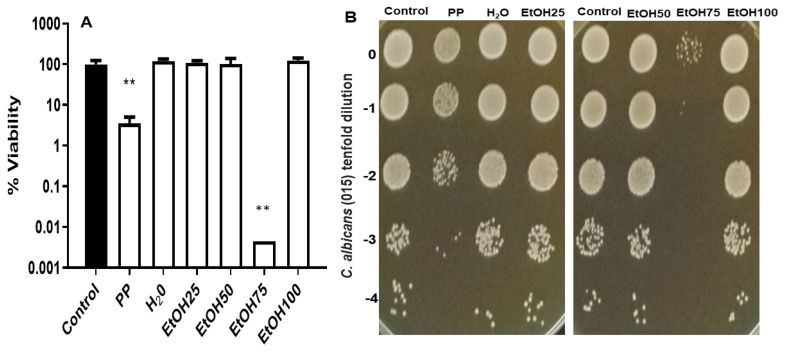
(**A**) Cell viability in liquid YPD medium of fungal exponential cultures treated with PP hydroethanolic fractions; statistically significant differences were ** *p* < 0.01 according to the Mann–Whitney U test. (**B**) Formation of macroscopic colonies on solid YPD plates at 37 °C of *C. albicans* 015 clinical strain upon exposure to the indicated fractions. The plates were scored after 48 h.

**Table 1 jof-09-00442-t001:** Quantification and yielding (%) of the raw PP fractionation after stepwise elution from Strata X hydrophobic cartridges with hydroalcoholic mixtures. Values are the mean of two experiments.

Fraction	Concentration (µg/mL)	Recovered mg (%)
Raw PP applied	500	100
H_2_O	12.55	2.51
EtOH25%	5.62	1.13
EtOH50%	21.38	4.28
TEtOH75%	124.33	24.87
EtOH100%	122.08	24.42

## Data Availability

Not applicable.

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
