# Peer review of "Insight into the Antifungal Effects of Propolis and Carnosic Acid—Extension to the Pathogenic Yeast Candida glabrata: New Propolis Fractionation and Potential Synergistic Applications"

_jof, 2023, doi:10.3390/jof9040442_

Round 1
Reviewer 1 Report
The keyword should be different than in title
What is the source of the isolates you used?
You did not mentioned anything about the statistical analysis in material and methods
The media composition and source should be mentioned
are you make results and discussion together if yes mentioned this in the results section
the discussion need to improve
as well as the conclusion
Author Response
Reply to Reviewer # 1
The criticisms of this reviewer mainly refer to questions on methodological defects and the need to improve important sections of the text. In addition, an important point regarding the statistical analysis has also been indicated. Because they are important for the presentation and scientific content of the manuscript, all of them have been incorporated in the revised version. Please, find our point-by-point answers.
The keyword should be different than in title
-Right. We have deleted the keywords contained in the title and added others more related to the species where the natural compounds are extracted.
What is the source of the isolates…?
-OK. Standard C. glabrata was obtained from the ATCC, whereas C. albicans was a clinical isolate from Hospital La Fe (Valencia, Spain). This has now been described in the text.
You did not mentioned anything about the statistical analysis….
-The reviewer is right, and we apologize for this omission. A new sentence is added in Methods: “The statistical significance of differences was determined by a Mann-Whitney U test” (p. 3, l. 100-101).
The media composition and source…
-Media were prepared in the laboratory by addition of individual components. The manufacturers are stated.
Are you make results and discussion together
-Thanks again. We involuntary made this mistake during the translation from word to the MDPI template.
The discussion need to improve…as well as the conclusion
-This question was also highlighted by Reviewer #3. Therefore, we have examined it with special detail. A new explanatory paragraph concerning discussion about possible mechanisms of action as well as hypothetical molecular target inside the cells has been introduced (p. 9, l. 324-345). Accordingly, the conclusion is also modified.
Reviewer 2 Report
Alternatives for antifungal therapies are important as the success rate depends on fungi, its strains, possible allergies in patient and number of other factors. Bloodstream infections as mentioned in line 154 are not only difficult to threat, but also difficult to diagnose. Thus, universal antifungal treatments (effective for several fungal pathogens) are important. However, this publication is dealing with fungi ex vivo and there is not given any indications, whether proposed propolis extract mixture with carnosic acid can be used perorally or ad usum externum.
Questions to authors:
There is given statistical variation of cell death values on Figure 1. It is expected to have similar description on Figure 2. Was there only one serial of experiment? How authors explain decrease of cell death in 5 hours with S (50:200) µg/mL and what was authors consideration not include this point – 5 hours in Figure 2?
Author Response
Reply to Reviewer # 2
We are grateful for all comments and suggestions raised by the reviewer. Notably, the majority focus on the potential clinical applications of our study. They have been addressed in the amended version of the manuscript according to the following point-by-point replies.
Alternatives for antifungal therapies are important as the success rate depends on fungi, its strains, possible allergies in patient and number of other factors. Bloodstream infections as mentioned in line 154 are not only difficult to threat, but also difficult to diagnose. Thus, universal antifungal treatments (effective for several fungal pathogens) are important...
-We undoubtedly agree with these comments, which are of application in clinical practice. Concerning the diagnosis and treatment, two patents for the oral use of CA+PP mixtures have already been registered. We are also performing a clinical trial in collaboration with a hospital. On the other hand, additional experiments should be carried out for the possible therapeutical use of the CA:PP mixtures as supplement of AmB supply in order to decrease the operative doses of this polyene. This approach would allow to minimize its undesired side-effects.
There is given statistical variation of cell death values…. Was there only one serial of experiment?
-These are certainly good remarks. In fact, we have already addressed this point in our answers to Reviewer #1. This query has been addressed in several ways: (i) A new Fig. 2 containing the corresponding error bars has been included; (ii) An indicative sentence in Methods: “The statistical significance of differences was determined by a Mann-Whitney U test” (p. 3, l. 100-101); (iii) The legend for Figures reports some statistical details.
How authors explain decrease of cell death in 5 hours with S (50:200) µg/mL …?
-Again, the reviewer is right. In fact, we observed that after long incubations, the medium adopted a brownish aspect, indicating possible alteration of some active principles in the CA:PP mixture. Therefore, the incubation was not prolonged beyond 2 h. A new sentence on this matter has been added (p. 4; l. 182-185).
Reviewer 3 Report
Authors utilize the combination mixture of carnosic acid and propolis to contain Candida glabrata growth as previously demonstrated for Candida albicans and Cryptococcus neoformans. Candida glabrata, an emerging pathogenic yests, displayed a lower susceptibility than Candida albicans and Cryptococcus neoformans. Authors also report how the pre-tretament of carnosic acid/propolis mixture enhances the antifungal effect of amphotericin B. The work is well presented and described. I would recommend to avoid picture frame on figures 1, 2, 4 and 5, to prepare a new figure 4 with a higher resolution and to re-format the figure 5. An important issue is the formulation of a possible mechanism of action of CA/PP mixture on the cell wall as hypothesis to corroborate by a molecular approach. What is the mixture molecular target? This aspect should be deeply investigated as the efficacy of the treatment and the differences among species could be due to the amount on the cell wall of the mixture molecular target.
Author Response
Reply to Reviewer # 3
The queries made by this reviewer deal with the significance of the data and the quality of the Figures. An important issue regarding the mechanism of action is also remarked. We truly appreciate all of them. In accordance, they have been attended in the amended version of the manuscript according to the following point-by-point replies.
Authors utilize the combination mixture of carnosic acid and propolis to contain Candida glabrata growth as previously demonstrated for Candida albicans and Cryptococcus neoformans. Candida glabrata, an emerging pathogenic yeasts, displayed a lower susceptibility… The work is well presented and described. I would recommend to avoid picture frame on figures 1, 2, 4 and 5, to prepare a new figure 4 with a higher resolution and to re-format the figure 5.
-The summary written by the reviewer is fully correct. Therefore, the frame of figures 1, 2, 4 and 5 has been eliminated and all the figures have been re-formatted to improve resolution. The size of letters in some labels has also been increased. Figure 4 is an original chromatogram, but according to your request, we have manually deleted the retention times of lower peaks for the sake of clarify. Retention times of the main peaks have been maintained. We hope these adjustments would be enough for the Reviewer’s requirements.
An important issue is the formulation of a possible mechanism of action of CA/PP mixture on the cell wall as hypothesis to corroborate by a molecular approach. What is the mixture molecular target?...
-This is clearly a crucial point. Indeed, the Reviewer #1 has come to the same conclusion. Although both Reviewers are right, we did not mention this point in the original version since up to this date there is not clear evidence about a conclusive mechanism(s) related to the antifungal action and the synergy observed among the different compounds tested, CA, PP and AmB. Anyway, we thank to the Reviewers for rising this point. As stated in our reply to Reviewer #1 an extensive and detailed paragraph analyzing possible mechanisms of action and molecular targets has been introduced at the end of results and discussion, which summarizes our present knowledge on this matter (p. 9, l. 324-345).
Round 2
Reviewer 3 Report
The manuscript has been improved with the required addition. Now the scientific soundness is complete and open some new aspects to be investigated. I would recommend the publication of the paper on JoF.
Author Response
We thank the prompt reply and positive evaluation made by this Reviewer.